# Tepary Bean (*Phaseolus acutifolius*) Lectins Induce Apoptosis and Cell Arrest in G0/G1 by P53(Ser46) Phosphorylation in Colon Cancer Cells

**DOI:** 10.3390/molecules25051021

**Published:** 2020-02-25

**Authors:** Ulisses Moreno-Celis, F. Josué López-Martínez, Ricardo Cervantes-Jiménez, Roberto Augusto Ferríz-Martínez, Alejandro Blanco-Labra, Teresa García-Gasca

**Affiliations:** 1Facultad de Ciencias Naturales, Universidad Autónoma de Querétaro, Av. de las Ciencias s/n. Juriquilla, Querétaro CP 76230, Mexico; ulisses.morenoc@gmail.com (U.M.-C.); bioxtremo@hotmail.com (F.J.L.-M.); ricardocervantesjimenez@gmail.com (R.C.-J.); raffer712701@gmail.com (R.A.F.-M.); 2Depto de Biotecnología y Bioquímica, Centro de Investigación y de estudios Avanzados del IPN-Unidad Irapuato, Guanajuato CP 36821, Mexico; alejandroblancolabra@gmail.com

**Keywords:** apoptosis, colon cancer, lectins, p53, *Phaseolus acutifolius*

## Abstract

A Tepary bean lectin fraction (TBLF) has been studied because it exhibits differential cytotoxic and anticancer effects on colon cancer. The present work focuses on the evaluation of the apoptotic mechanism of action on colon cancer cells. Initially, lethal concentrations (LC_50_) were obtained for the three studied cell lines (HT-29, RKO and SW-480). HT-29 showed the highest LC_50_, 10 and 100 times higher than that of RKO and SW-480 cells, respectively. Apoptosis was evaluated by flow cytometry, where HT-29 cells showed the highest levels of early and total apoptosis, caspases activity was confirmed and necrosis was discarded. The effect on cell cycle arrest was shown in the G0/G1 phase. Specific apoptosis-related gene expression was determined, where an increase in p53 and a decrease in Bcl-2 were observed. Expression of p53 gene showed the maximum level at 8 h with an important decrease at 12 and 24 h, also the phosphorylated p53(ser46) increased at 8 h. Our results show that TBLF induces apoptosis in colon cancer cells by p-p53(ser46) involvement. Further studies will focus on studying the specific signal transduction pathway.

## 1. Introduction

The development of new therapies against colon cancer is gaining great importance in the research of natural products. Among plant compounds that have been studied because of their anticancer effects, lectins are being strongly considered. They are glycoproteins with the ability to recognize specific cell membrane carbohydrates [1,2]. Some plant lectins have shown differential cytotoxic effects on cancer cell lines given the specific changes in cell membrane glycosylation patterns [3,4]. The cytotoxic effects of plant lectins on cancer cell lines are related to their high specificity to cell membrane carbohydrates [5].

The specificity on different tumor cell lines could reflect different progression stages [6]. Some lectins can be internalized, causing cell death through ribosomal inactivation, or they can initiate signaling cascades that lead to apoptosis [5,7]. The most recognized mechanisms of action are the following: (1) At a physiological level, lectin-lymphocyte binding has been observed as well as release of blood cytokines, activation and release of spleen lymphocytes, activation of NK cells and macrophages, production of antiangiogenic factors, combination of intestinal hyperplasia and antiangiogenic effects reducing nutrient availability and cytotoxic effects on tumor cells [8,9]. (2) At the biochemical and molecular level, different mechanisms of action are proposed. One mechanism describes the binding of lectins to surface adhesion molecules that participate in a wide variety of transduction signals important for cell regulation. A second mechanism suggests that lectins affect the fundamental cellular process for cell division [10,11]. (3) A third level explains that lectin induces apoptosis in several ways: dependent on the intracellular activation of caspase-8/FLICE, the activation of caspase-3 and Poly (ADP-ribose) Polymerases (PARP) cleavage, the activation of Bax (pro-apoptotic protein) and the inhibition of both Bcl-2 (apoptosis suppressor) and telomerase [5,11,12,13,14]. It is important to note that it is not an indispensable requirement for lectin to be internalized, since apoptotic effects can be caused by the interaction of lectin with cell membrane receptors.

Some lectins have been studied against colon cancer because of their effects on cell growth and promotion of cell death in different cell lines [13,15,16], such as lectins of *Arisaema helleborifolium* (AHL), *Arisaema tortuosum* (ATL), *Arachis hypogaea* (PNA), *Viscum album* (VAL, VAA, VAA-1), *Sauromatum venosum* (SVA), *Phaseolus vulgaris* L. (PVA) and *Vicia faba* (VFA) [4,5,6,8,14,15,17]. A Tepary bean (*Phaseolus acutifolius*) lectin fraction (TBLF) has shown differential cytotoxic effects on breast, cervix and colon cancer cell lines as well as on non-malignant cells from the intestinal tract [15,18]. Similar effects were observed with other Tepary bean lectin fractions on different colon cancer cell lines [19,20]. In vivo studies have reported that TBLF (50 mg/kg by intragastric administration for six weeks every third day) exhibits low toxicity, good tolerability and activates the immune system [21,22]. Between the observed adverse effects are loss of body weight, small intestinal villus and colonic crypt atrophy and exocrine pancreas hypertrophy, without systemic adverse effects. Negative effects could be reversible after a recovery period [8]. When TBLF was tested against colon cancer (chemically induced using dimethylhydrazine (DMH) or azoxymethane (AOM) in rats), early tumorigenesis inhibition was linked with modulation of apoptotic pathways [23]. The present work focuses on the evaluation of the apoptotic effect of TBLF on colon cancer cells, mainly by studying molecular mechanisms.

## 2. Results

### 2.1. Concentration-Response Study and LC_50_ Determination

Concentration-response curves were obtained for each cell line, the TBLF-LC_50_ values were 402 µg/mL for HT-29 cells, 49.2 µg/mL for RKO cells and 4.7 μg/mL for SW-480 cells (Figure 1). Our results suggest that TBLF is able to recognize cancer cells in a differential manner, provoking specific cytotoxic effects, even in cells derived from the same pathology. This differential effect was previously demonstrated for TBLF [15,24]. These results show that HT-29 cells were almost 10 times and 100 times more resistant than RKO and SW-480 cells, respectively. One of the relevant differences between the cell lines was the expression of EGFR for SW-480 cells, which was the most sensitive cell line to TBLF effects [5,6,13,14].

### 2.2. Effects on Cell Death and Cell Cycle Arrest 

Apoptosis induction by TBLF was confirmed by annexin V determination (*p* ≤ 0.05) using the LC_50_ for each cell line (Figure 2). A decrease in cell viability was determined in the three cell lines with respect to control cells (*p* ≤ 0.05). Early apoptosis was observed with a 21.7% increase in HT-29 cells, 15% in SW-480 cells and 3% in RKO cells after 8 h treatment; late apoptosis had a 1% increase in HT-29 cells, 7% in SW-480 cells and 25% in RKO cells. Total apoptosis (subtracting baseline apoptosis in control cells) was 22.77% for HT-29 cells, 23.3% for RKO cells and 18.31% for SW-480 cells. Differential effects were observed again and the apoptosis mechanism was determined in HT-29 cells because this cell line showed the highest level of early and total apoptosis.

The cytotoxic effect of TBLF was tested (Figure 3), where no necrotic effect after treatment with TBLF-LC_50_ for 8 h was observed. Several studies have shown that induction of apoptosis by the activation of multiple caspases is a common mechanism of various lectins [25]. Caspase-3, an apoptosis effector protein, is currently considered a marker of this process [26]. In the present work, increases of 30% of caspase-3 activity and 50% of total caspases activity were observed with respect to control cells (*p* ≤ 0.05) after 8 h treatment with TBLF-LC_50_. Cell cycle arrest showed an increase of 27.4% in the G0/G1 phase with respect to the negative control (*p* ≤ 0.05) (Figure 4), but no effect was observed in S and in G2/M phases.

### 2.3. Apoptotic-Related Gene Expression and Phosphorylation of P53 in Ser46 

Significant changes in apoptotic gene expression were observed after TBLF-LC_50_ treatment (Figure 5). A decrease in the expression of Bcl2 and an increase in p53 were determined, suggesting that TBLF mainly affected the anti-apoptotic pathways. Changes in p53 expression from 0 to 24 h showed and increase between 4 to 8 h with a significant decrease at 12 to 24 h. Phosphorylation of p-p53(ser46) showed an increase, particularly during the first 8 h and subsequently was maintained. These results suggest that the specific activation effect of p53(ser46) is related to an increase of p53 gene expression, where the apoptotic signal is carried out.

## 3. Discussion

Previous studies have shown that TBLF exhibits differential cytotoxic effects on cancer cell lines [15]. In an in vivo study, this lectin fraction inhibited early malignant lesions in the colon when rats were treated with AOM as a cancer inductor and apoptosis was associated with a decrease in phosphorilated form of Akt (p-Akt) [23]. In order to know the apoptotic mechanism of action, TBLF was tested in three different colon cancer cell lines. LC_50_ values were calculated and the results showed that HT-29 adenocarcinoma cells exhibited the most resistant phenotype, with an LC_50_ almost 10 and 100 times higher than that of RKO and SW-480 cells, respectively. HT-29 and RKO cells express a urokinase receptor, while SW-480 cells are positive for the epidermal growth factor receptor (EGFR). As SW-480 cells exhibited the lowest LC_50_, suggesting that they were the most sensitive cells and the fact that in the in vivo study a decrease in p-Akt was observed, it is probably that the EGFR may be involved in lectin-cell interactions [23]. The mechanisms reported for cytotoxic effects of plant lectins have been described in a wide range of cellular or physiological events such as activation of the immune system, induction of apoptosis by blocking membrane receptors, provoking mitochondrial imbalance and effects on some signal transduction pathways [1,4,27]. HT-29 cells showed the highest level of early and total apoptosis, while SW-480 cells exhibited the higher level of late apoptosis; this result agrees with the fact of the sensitiveness of SW-480 cells.

As HT-29 cells showed the highest levels of early and total apoptosis, the mechanism of action was evaluated using this cell line. Necrosis was discarded and this result agrees with the observed effects for other lectins, such as *Phaseolus coccineus* lectins, where it was determined that it achieves only an 11% LDH increase after 24 h on L-929 fibroblasts [28]. In the same way, *Clematis montana* lectins were tested on L929 cells and gave very similar results to those reported for *Phaseolus coccineus* lectins [29]. Apoptosis was determined by total caspases and caspase-3 activities, where TBLF increased both parameters by 50% and 30%, respectively.

Apoptosis has been observed for other lectins such as *Morus alba* L., mulberry leaf lectins (MLL), where caspase-3 activity was increased in breast (MCF-7) and colon (HCT-15) cells in a similar way to cisplatin [11]. In the same way, mistletoe (*Viscum album* L.) lectins (VLL) induced caspase-3 activation in *all* leukemia cells [12] and similar effects were observed for lectins of the Chinese medicinal plant *Astragalus membranaceus* on K562 leukemia cells. Concanavalin A (Con A) and *Sophora flavescens* lectins (SFL) induced caspase-mediated cell death and caspase-3 activity in MCF-7 breast cancer cells in a concentration-dependent manner [13]. In the same way, the induction of apoptosis may have been orchestrated by mediating proteins such as NF-κB, p73, Akt and p53, as well as by the processes of autophagy and oxidative stress [25].

Cell cycle showed arrest in the G0/G1 phase after TBLF treatment for 8 h in HT-29 cells. Such a result agrees with several studies that have shown the effect of lectins on cell cycle. An aqueous extract of *Viscum articulatum Burm*. F. (VAQE) stopped the cell cycle in G2/M in human leukemia cells and a *Moringa oleifera* seed lectin (MOSL) inhibited cell growth by arresting cell cycle in G2/M in Ehrlich–Lettre ascites carcinoma (EAC) cells [30]. Specifically, in the metastatic SW-620 colon cancer cell line, arrest was found in the S phase and was time-dependent. Arrest was also induced by *Rhizoctonia bataticola* lectin (RBL) on cells with chromosomic structural loss [31]. Pea lectins provoke differential cell cycle arrest, dependent on the cell line, causing G2/M arrest in SW-48 and G0/G1 arrest in SW-480 cells, both from human colon cancer [32].

Shi et al., 2014 observed a dose-dependent effect of Con A lectin on the apoptotic promoter genes’ Bid and Bax expression, proteins from the same family as Bad. However, it was observed that the expression of Bad did not show differences with respect to the control. Bad protein can be induced by Akt dephosphorylation, which has been observed to be affected by lectins of mistletoe VCL, in turn inducing apoptosis [17,33]. In the present work, a decrease in Bcl2 expression and an increase in p53 was determined. In the previous in vivo studies using TBLF, premalignant lesions in the colon were induced with AOM [23], but, in that moment, it was not possible to determine the effect of TBLF on p53 because AOM induction causes apoptosis-induced proliferation (AiP) that increases the p53 level [34,35].

Multiple signaling pathways having different activators and deactivators, as well as changes in its own pathway regulation, can affect the activity of p53. The participation of p53 as a transcription factor and as a tumor suppressor is very important in making decisions for the fate of a damaged cell, although the mechanisms by which it can be induced are not fully known [36,37]. Specific phosphorylation has been observed in various events of death and cell survival that trigger the specific activation of genes or regulate the permeability of the mitochondrial membrane. It has been observed that phosphorylation of p53 in ser15 and ser20 occur during slight damage to DNA, while phosphorylation in ser46 is involved with cell death [37,38,39]. Phosphorylation of p53 in ser46 is related to genotoxic stress and occurs after several hours of cell damage when the cell death process could be considered irreversible, which leads to cell cycle arrest and activation of control points [39,40]. It has been seen that some natural compounds promote the phosphorylation of p53 in ser46; such is the case of the induction of cell death by an extract of *Zelkova serrata*, which arrests the cell cycle in S-phase, activation of caspase-8 and an increase in the amount of p-p53(ser46) in oral cancer cells in contrast to non-cancerous fibroblasts [41]. It has been observed that there is a genotoxic effect dependent on the activity of p-p53(ser46) induced by treatment with quercetin and curcumin [42].

In this sense, when p53 is phosphorylated in serine 46, it induces signaling pathways triggering apoptosis by stopping the cell cycle [37,43,44]. Our results suggest that p53 mutations in HT-29 cells do not affect the phosphorylation in ser46 and TBLF treatment increases the p-p53(ser46) level at 8 h. Similar results have been observed in various studies for lectin-mediated cell death induction pathways [27,45,46,47,48]. Some other lectins affect the p53-mediated apoptosis pathway [28,49,50].

## 4. Materials and Methods

### 4.1. Obtaining TBLF

Tepary bean (*Phaseolus acutifolius*) seeds were obtained from a local market in Hermosillo, Sonora, Mexico. A sample of Tepary bean was deposited and identified in the herbarium of Dr. Jerzy Rzedowski of the Natural Sciences Faculty, Querétaro Autonomous University, Santiago de Querétaro, México. The TBLF was obtained as described previously [51] with modifications [15,52]. Briefly, bean seeds were ground and degreased using a methanol/chloroform 2:1 solution, subsequently an aqueous extract was obtained using Tris buffer pH 6.8, precipitated with ammonium sulphate (40 to 70% saturation), centrifuged, dialyzed and separated by size-exclusion chromatography (Sephadex G75 column). Protein quantification was obtained by the Bradford method [53] and agglutination was tested [54] using glutaraldehyde-treated erythrocytes [55].

### 4.2. Cell Culture and Concentration–Response Assay

Colon cancer cells HT-29, RKO and SW-480 were obtained from the American Type Culture Collection (ATCC^®^) (Table 1) [56,57]. Cells were seeded in 60 mm diameter dishes with Dulbecco’s modified Eagle’s medium, DMEM, (GIBCO, New York, NY, USA), supplemented with 10% fetal bovine serum (FBS, Biowest, Nuaillé, France) at 37 °C under a 5% CO_2_ saturated water atmosphere, with medium changes every two days until confluence.

Cytotoxic effects were determined by seeding 3 × 10^4^ cells per well in 24-well dishes in DMEM medium with 10% FBS for 48 h. Then they were synchronized with DMEM at 2% FBS for 24 h and different concentrations of TBLF were added (HT-29 cells: 1, 10, 100, 500, 1000 and 1500 µg/mL; RKO cells: 1, 5, 10, 20, 40, 80, 200, 280, 320 and 400 µg/mL; and SW-480 cells: 1, 5, 10, 20, 100 and 200 µg/mL) in DMEM/0.5% FBS for 24 h. The cells were collected after a 5 min incubation in trypsin/EDTA (0.15 Mm/0.5 M)/ and a direct count was performed with a Newbauer chamber (Merk (BRAND®), Darmstadt, Germany). Cell number was determined following the formula (1) and the lethal concentrations (LC_50_) were obtained by simple linear regression using the concentration log_10_ vs. survival percent.
(1)$$Total\;cells=\frac{\left(Number\;of\;cells\;counted\right)}{\left(Number\;of\;fields\right)}\times 10,000\times mL\;Suspension$$


### 4.3. Evaluation of Apoptosis by Flow Cytometry

Cell death evaluation was performed using the Muse^®^ Annexin V and Dead Cell Assay Kit (Milipore cal. No: MCH 100105, Darmstadt, Germany) for the three cell lines. Briefly, cultures were maintained under the previous described conditions until they reached 70% confluence. Three groups were formed: negative control, incubated with 0.5% bovine serum albumin (BSA) in DMEM; treated group, incubated with the TBLF LC_50_ (HT-29, 402 μg/mL; RKO, 49 μg/mL; SW-480, 4.7 μg/mL) in 0.5% BSA DMEM; and a positive control, treated with camptothecin 5 μM in DMEM 0.5% BSA for 8 h. Camptothecin is an antineoplastic alkaloid extracted from the Chinese tree *Camptotheca acuminate*. Its mechanism of action is recognized for cell cycle arrest in the S phase, for promoting disruptions in the double strand of DNA and for stabilizing the topoisomerase I-DNA complex promoting apoptosis [57]. Cells were collected by trypsinization and concentrated by centrifugation (6000× *g* for 5 min), then they were washed with 1 Mm PBS to disaggregate the clustered cells. Cells in each group were adjusted to 1 × 10^6^ cells per mL in the culture medium and determinations were done following the supplier indications. The experiments were performed in triplicate in at least two independent experiments.

### 4.4. Necrosis Determination by Lactate Dehydrogenase Assay

Lactate dehydrogenase (LDH) is used as a marker of cell necrosis. HT-29 cells were cultured as previously described in 24-well plates and divided into three groups by triplicate: negative control, incubated with 0.5% BSA in DMEM for 8 h at 37 °C; TBLF treated group, incubated with the LC_50_ (402 μg/mL) in 0.5% BSA-DMEM for 8 h at 37 °C; and a positive control, treated with 1% Triton 100X (JT Baker, cat No. X198-07, Madrid, Spain) in 0.5% BSA-DMEM and incubated at 37 °C for 30 min. The conditioned media were obtained and the LDH-Cytotoxicity Assay kit (Biovision, cat No. K311-400, Milpitas, CA, USA) was used to evaluate the necrotic effect according to the manufacturer’s instructions. Samples were read at 492 nm and the cytotoxicity percentage was calculated based on the enzymatic activity of LDH, according to the following formula (2):
(2)$$\%\;\mathrm{Cytotoxicit}=\mathrm{Treatment}-\text{Negative Control}\times \frac{100}{\text{Positive Control}}-\text{Negative Control}$$


### 4.5. Caspase- 3 Activity by Colorimetric Assay and Flow Cytometry Multi-Caspase Assay

Caspase-3 activity was determined using the Caspase-3/CPP32 Colorimetric assay kit (BioVision, Milpitas, CA, USA) according to the manufacturer’s protocol. HT-29 cells were cultured in 30 mm plates (3 × 10^4^ cells/plate) for 8 h and then LC_50_ of TBLF (402 μg/mL) was added to each well. After incubation for 8 h, cells were harvested, washed with phosphate-buffered saline (PBS) and suspended in cold lysis buffer. The cells were placed on ice for 20 min and lysed cells were centrifuged at 14,000× *g* for 15 min. For the caspase assay, samples in assay buffer were mixed with caspase substrate (Ac-DEVD-pNA) in a 96-well plate. After overnight incubation at 37 °C, the absorbance of released p-nitroaniline was measured at 405 nm using a microplate reader (Spectra MAX 250, Molecular Devices). Caspase-3 activity was determined as follows (3):
(3)$$\mathrm{Caspase}-3\;\mathrm{activity}=\frac{\mathrm{Absorbance}}{µg\;\mathrm{of}\;\mathrm{total}\;\mathrm{Protain}}/\frac{\text{cells number}}{µg\;\mathrm{of}\;\mathrm{total}\;\mathrm{Protain}}$$


The flow cytometry multi-Caspases assay (for caspases −1, −3, −4, −5, −6, −7, −8 and −9) was performed according the apoptosis assay by annexin V determination, as described previously, using the Muse^®^ MultiCaspase- Kit (Cat. No. MCH100109, Merk Millipore Inc, Darmstadt, Germany). Briefly, HT-29 cells were incubated with LC_50_ for 8 h, harvested and incubated with 5 μL of Muse multi-caspase reagent working solution at 37 °C for 30 min. After the incubation, 150 μL of Muse Caspase 7-AAD working solution was added to each sample. The activities of caspases were determined with the Muse™ Cell Analyzer (Merck Millipore, Darmstadt, Germany).

### 4.6. Cell Cycle Analysis

Flow cytometry was performed by seeding 3 × 10^5^ cells in 60 mm culture dishes in 10% FBS-DMEM and incubating them at 37 °C as previously described. When cultures reached 70% confluence, they were divided into three groups per triplicate: negative control, which was incubated in 0.5% BSA-DMEM; TBLF treated cells, added with TBLF LC_50_ in 0.5% BSA-DMEM; and a positive control, using 5 μM Camptothecin in 0.5% BSA-DMEM. All groups were incubated for 8 h and cells were collected by trypsinization and centrifugation (6000× *g* for 5 min). Cells were washed with 1x PBS/1 mM EDTA to disaggregate cells and then fixed in 70% ethanol for 4 h at −20 °C following the instructions (Muse^®^ Cell Cycle Assay Kit, Merk Millipore. Cat. No. MCH1006, Darmstadt, Germany). All assays were made in triplicate and with at least two independent experiments.

### 4.7. Gene Expression Evaluation

Treated cells were cultured as previously indicated for the flow cytometry assay. RNA extraction and purification was carried out by adding 400 μL of Trizol (Invitrogen ™, Carlsbad, CA, USA) to cells in 60 mm plates, homogenizing and kit instructions were followed (Direct-zol™ RNA MiniPrep RNA extraction kit was used, Zymo Research, Cat. No. R2052, Irvine, CA, USA). Samples were resuspended in nuclease-free water, total RNA was quantified and purity was determined by spectrophotometry using a NanoDrop™ 2000/2000c spectrophotometer (Thermo Scientific, Waltham, MA, USA). cDNA synthesis was achieved for each 2 μg of RNA added in 200 μL microtubes and following the supplier’s instructions for the kit (Maxima H Minus First Strand cDNA Synthesis, Thermo Scientific. Cat. No. K1652, Waltham, MA, USA) for a reaction volume of 40 μL and a cycle of 25 °C for 5 min, 65 °C for 30 min and 85 °C for 15 min. 

Primers were designed selecting genes related to the colorectal carcinogenesis signaling pathway. Genetic sequences were analyzed by the UCSC Genome Browser of the University of California [58] and primers were made by using the Primer3 page [59], seeking a TM of 60 ± 2 °C (for 20 ± 2 bp), a product size of 100–250 bp and CG ≥ 50%. Sequences were synthesized by Sigma Laboratories (Aldrich, Mexico) (Table 2). The q-PCR reaction was performed in 96-well PCR plates using 3.4 μL of nuclease-free water, 5 μL of SYBR^®^ Select Master Mix for CFX (Applied Biosystems, Cat. No. 4472942, Foster City, CA, USA) and 1 μL of cDNA template for each reaction. A BioRad thermocycler (CFX96 model C1000, Bio-Rad Laboratories, Inc, Hercules, California, USA) was used with the following conditions: 95 °C for 10 min (15 s at 95 °C, 30 s at 60 °C, 30 s at 72 °C) for 35 cycles and then held at 16 °C until determination was done.

### 4.8. Protein Determination by Western Blot

Total protein was extracted from a pellet of approximately 5 × 10^6^ HT-29 cells per time period (2, 4, 8 and 12 h) from the two groups: negative control (0.5% BSA-DMEM) and TBLF LC_50_ (0.0402 mg/mL). Cells were lysed using the CelLytic™ MT Cell Lysis Reagent (Sigma Aldrich, Cat. No. C3228, St. Louis, MO, USA) and total protein was determined [53]. Protein concentration was adjusted to 5 µg/µL. The samples were boiled for 5 min, separated on a 12% acrylamide (30%)/bis-acrylamide (1%) SDS-PAGE gel [60] and transferred to a 0.2 µm pore nitrocellulose membrane (Bio-Rad Laboratories, Inc Hercules, California, CA, USA). Membranes were subsequently blocked with Blotting-Grade Blocker (Cat. No. 1706404, Bio-Rad Laboratories, Inc, Hercules, California, CA, USA) for 2 h, incubated for 12 h with the primary rabbit antibody p53 (Cat. No. sc-98) or p-p53 (ser46) (Cat. No. sc-101764, Santa Cruz antibodies). The membrane was then washed (1X TBS/1% Tween 20), incubated with the secondary antibody (Goat anti-rabbit IgG, Cat. No. PI-1000, Vector Laboratories) for 4 h and finally revealed using Amersham ECL Detection Reagents (Cat. No. RPN2105, General Electric, Boston, MA, USA). The images obtained after development were analyzed with the ImageJ^®^ program. All assays were made in triplicate and with at least two independent experiments.

### 4.9. Statistics

Results were analyzed using the SPSS v.26 program. For comparison for two groups a Student *t* test and ANOVA one-way for three or more groups were used (*p* ≤ 0.05). Data were plotted using Prism Graph v.6 (La Jolla, CA, USA) comparing the means of each group ± standard deviation. Western blot images were analyzed with the ImageJ program to de-merge the density of the bands.

## 5. Conclusions

The results indicate that TBLF induces death in colon cancer cell lines in a dose- and cell line-dependent manner. In HT-29 cells, TBLF induced G0/G1 arrest and apoptosis induction was determined by activation of caspases, particularly caspase-3 and by flow cytometry. An increase in phosphorylation of p53 in serine 46, which is highly involved to the apoptotic process, was observed. Further works will focus on studying the effects of the signal transduction pathway, especially in the relationship between EGFR-Akt pathway and induction of apoptosis.

## Figures and Tables

**Figure 1 molecules-25-01021-f001:**
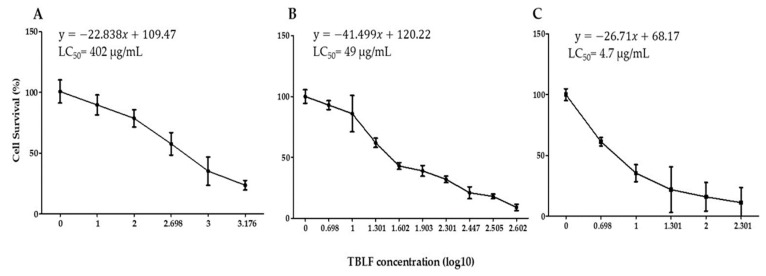
Concentration–response curves for HT-29, RKO and SW-480 cells. Cells were treated with different concentrations of Tepary bean lectin fraction (TBLF) for 24 h. (**A**) HT-29 cells (1, 10, 100, 500, 1000 and 1500 µg/mL). (**B**) RKO cells (1, 5, 10, 20, 40, 80, 200, 280, 320 and 400 µg/mL). (**C**) SW-480 cells (1, 5, 10, 20, 100 and 200 µg/mL). A linear regression of log10 of the TBLF concentration vs. cell survival (%) was calculated in each case.

**Figure 2 molecules-25-01021-f002:**
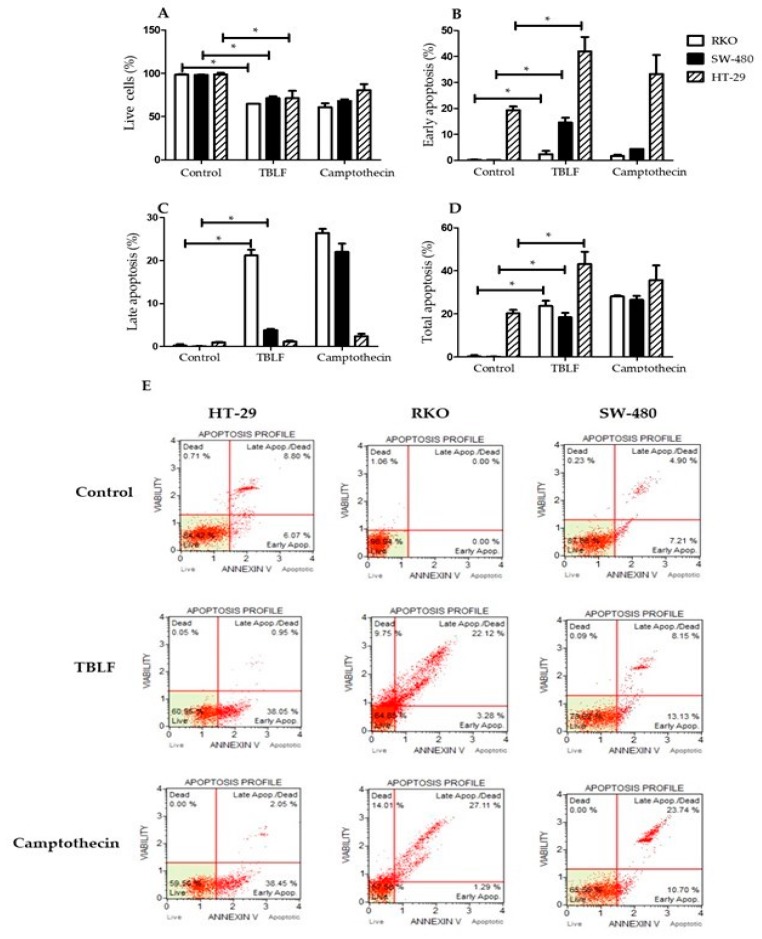
TBLF effect on apoptosis induction. Cells were treated for 8 h with the lethal concentration (LC_50_). (**A**) Live cells, (**B**) early apoptosis, (**C**) late apoptosis, (**D**) total apoptosis. Camptothecin (5 μM) was used as a positive control and 0.5% bovine serum albumin (BSA) as a negative control. (**E**) Flow cytometry representative dot plots are shown. (*) Statistically significant difference (Student *t* test, *p* ≤ 0.05).

**Figure 3 molecules-25-01021-f003:**
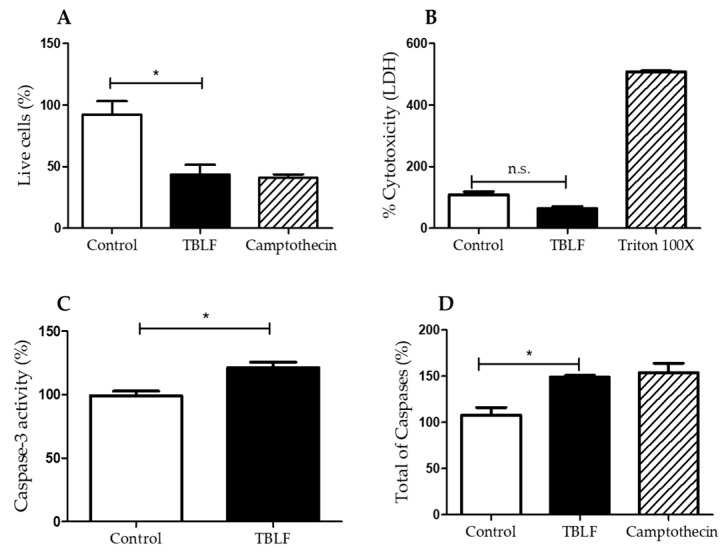
Effect of TBLF on necrosis and activation of caspases in HT-29 colon cancer cells. Cells were treated with the TBLF-LC_50_ for 8 h. (**A**) Cell viability (live cells), (**B**) lactate dehydrogenase release as necrosis marker, (**C**) caspase-3 activity, (**D**) total caspases activity. Camptothecin (5 μM) was used as a positive control and 0.5% BSA as a negative control. (*) Statistically significant difference (Student *t* test, *p* < 0.05).

**Figure 4 molecules-25-01021-f004:**
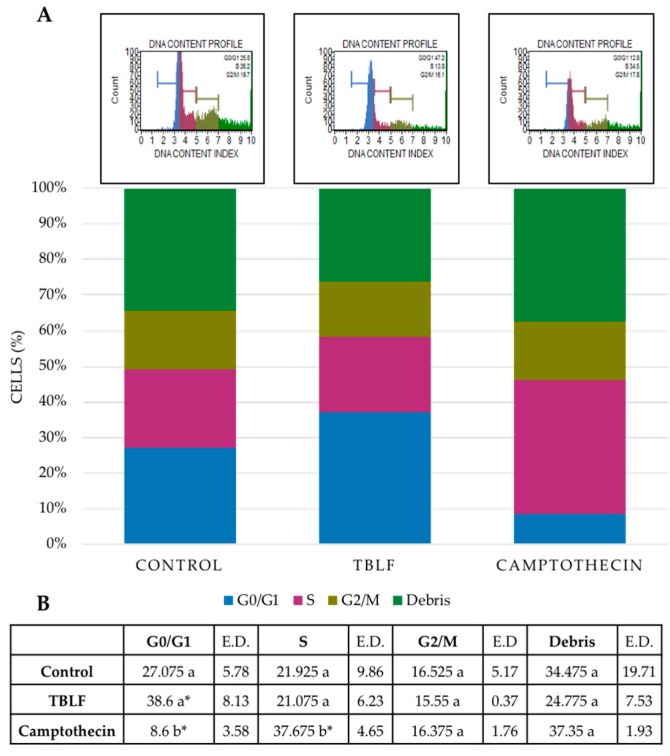
Effect of TBLF on cell cycle arrest on HT-29 colon cancer cells. Cells were treated with the TBLF-LC_50_ for 8 h. (**A**) Representative results of the cell cycle analysis; control group (BSA 0.5%), TBLF-LC_50_ and positive control camptothecin (5 µM). (**B**) Graphic results obtained in the cell cycle analysis. One-way ANOVA was performed for each cell cycle phase. Small letters indicate significant differences (Tukey *p* ≤ 0.05). (*) Indicates significant difference (Dunnett *p* ≤ 0.05) with respect to the negative control group.

**Figure 5 molecules-25-01021-f005:**
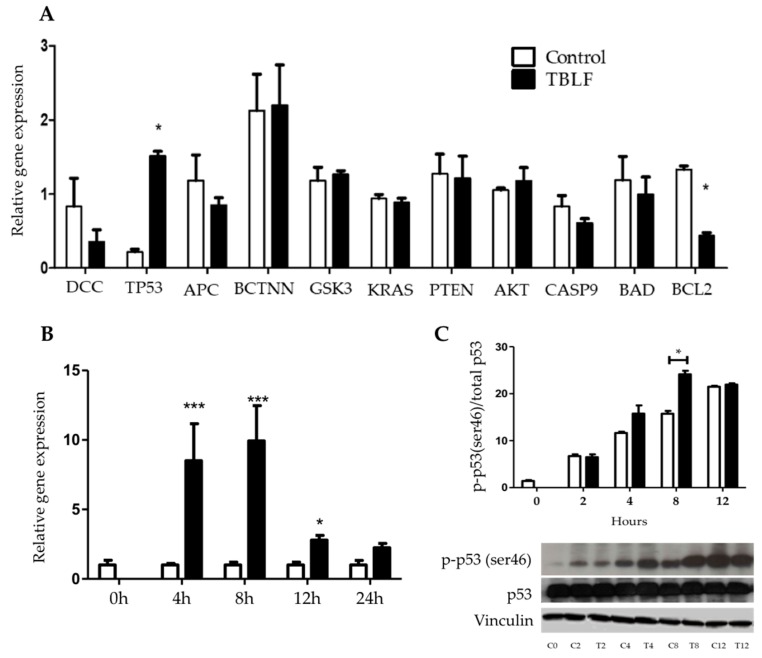
Effect of TBLF-LC_50_ on apoptosis and cancer-signaling pathway gene expression in HT-29 colon cancer cells. (**A**) Cells were treated with the LC_50_ of TBLF for 8 h. Relative gene expression for DCC, P53, APC, BCTNN, GSK-3, KRAS, PTEN, AKT, CAS9, BAD and BCL-2 with respect to B-actin. (**B**) Cells were treated with the TBLF-LC_50_ for 0, 4, 8, 12 and 24 h. RT-qPCR evaluation for total p-53 gene respect to B-actin. (**C**) Western blot for total p53 and p-p53(ser46), graphic results are presented as p-p53(ser46)/total p-53. Student t test significant difference: (*) *p* ≤ 0.05 and (***) *p* ≤ 0.001.

**Table 1 molecules-25-01021-t001:** Studied colon cancer cell lines.

Cell Line	Characteristics
HT-29 (ATCC® HTB-38™)	Colorectal adenocarcinoma cells. Positive for c-myc, K-ras, H-ras, N-ras, Myb, sis and fos oncogenes expression, N-myc oncogene expression was not detected. p53 protein is overproduced with a G-A mutation in codon 273 resulting in an Arg-His substitution. Express human adrenergic alpha2A receptor, urokinase receptor (u-PAR) and moderate expression of vitamin D receptor
RKO (ATCC® CRL-2577™)	Colon carcinoma cells. Wild-type p53 with high expression, positive for urokinase receptor (u-PAR), but lack endogenous human thyroid receptor nuclear receptor (h-TRbeta1).
SW-480 (ATCC® CCL-228™)	Colorectal adenocarcinoma cells. Positive for the expression of c-myc, K-ras, H-ras, N-ras, myb, sis and fos oncogenes, negative for N-myc oncogene expression and for Matrilysin (a metalloproteinase associated with tumor invasiveness), express high levels of p53 protein with a G-A mutation in codon 273 of the p53 gene resulting in an Arg-His substitution and a C-T mutation in codon 309 resulting in a Pro-Ser substitution. Positive for epidermal growth factor receptor (EGFR).

**Table 2 molecules-25-01021-t002:** Primer sequences for Real-Time Polymerase Chain Reaction (qPCR).

Gen Target	Forward	Reverse
**B-CTNN**	TGGACTTGATATTGGTGCCCA	GCCACCCATCTCATGTTCCA
**DCC**	CCCCTGAAGTGTCTGAGGAG	AGCTGCTTCATGAGTCCTTCC
**PI3K**	TGGAGCTGACCCAAATCCAT	TTCAAAGGCAGGGTTACTCC
**GSK3**	CTCCATCCAACCGTCTCTCA	GGTAGGTGTGGCATCGGTC
**CAS9**	CAAGAGTGGCTCCTGGTACG	TCCCTTTCACCGAAACAGCA
**BAD**	TTCGGAGGATGAGTGACGAG	CAAGTTCCGATCCCACCAGG
**PTEN**	GCCGTCAAATCCAGAGGCTA	GGATCAGAGTCAGTGGTGTCA
**AKT**	CCTTCAAGCCCCAGGTCAC	CGCTCGCTGTCCACACAC
**TP53**	CCAACAACACCAGCTCCTCT	TCAGGAAGTAACACCATCGTAAG
**BCL-2**	GACTGAGTACCTGAACCGGC	GGCCAAACTGAGCAGAGTCT
**KRAS**	TGTGATTTGCCTTCTAGAACAGT	ACACCCTGTCTTGTCTTTGCT

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
