# Peer review of "Tepary Bean (*Phaseolus acutifolius*) Lectins Induce Apoptosis and Cell Arrest in G0/G1 by P53(Ser46) Phosphorylation in Colon Cancer Cells"

_molecules, 2020, doi:10.3390/molecules25051021_

Round 1
Reviewer 1 Report
There are English writing issues throughout the manuscript.
A few examples: "mama" cells rather than "mammary"; "cytotoxicity" is spelled wrong in Figure 3; "caspase" is used in some areas, but "caspasa" in others.
Figure 2. How are early and late apoptosis distinguished and what is the significance of quantifying it this way?
Figure 3. In the text, it is stated that caspase 3 and 8 activity are measured, how are these activities measured?
What is the significance of the small cell cycle data?
Author Response
Comments and Suggestions for Authors
There are English writing issues throughout the manuscript.A few examples: "mama" cells rather than "mammary"; "cytotoxicity" is spelled wrong in Figure 3; "caspase" is used in some areas, but "caspasa" in others.
A: Thank you very much. Spelling mistakes have been corrected and the document has been sent for language revision.
Figure 2. How are early and late apoptosis distinguished and what is the significance of quantifying it this way?A: Apoptosis can be detected by using markers for phospahtidylserine membrane translocation for early stages of apoptosis and DNA fragmentation for latter stages of apoptosis. In this case, Figure 2 shows representative dot plots by the flow cytometry analysis where phosphatidylserine was detected using annexin V (“x” axis) and DNA fragmentation with, 7-Aminoactinomycin D (7-AAD) (“y” axis). Four different stages of cell viability are distinguished [1]. 1) Low signal of both, annexin V and 7-ADD means live cells (lower left panel), 2) high annexin V and low 7-ADD means early apoptosis (lower right panel), 3) high annexin V and high 7-ADD means latter apoptosis (upper right panel), and 4) low annexin V and high 7-ADD means fully dead cells, mostly nuclear debris (upper left panel); each panel was represented in the bar graphs. Quantifying apoptosis by this way shows which cell line is more sensitive to the treatment than another, as well if the treatment is being effective to provoke apoptosis. In that sense, when using the calculated TBLF LC50for each cell line for 8 h of treatment, we can observe a similar decreased live cells percent in all cases. RKO cells showed the highest percent of latter apoptosis, suggesting that the TBLF exhibits an acute effect on cell death, while HT-29 cells showed the highest percent of cells in early and total apoptosis suggesting that this cell line is more resistant to the TBLF treatment than RKO cells. For SW-480 cells, level of early apoptosis was of 15% and late apoptosis of 7%, suggesting that the effect at 8 h is probably not enough at this treatment time. Our results show that RKO cells were more sensitive than HT-29 cells because apoptosis was in an advanced stage. Camptothecin was used as a positive control for apoptosis.
http://www.icms.qmul.ac.uk/flowcytometry/uses/musekits/protocols/MCH100105%204600-3384MAN%20[B]%20%20ANNEXIN%20V%20&%20DEAD%20CELL%20KIT%20100%20TEST%20USER%27S%20GUIDE.pdf Figure 3. In the text, it is stated that caspase 3 and 8 activity are measured, how are these activities measured?A: Figure 3 shows caspase-3 activity (Figure 3C) and total caspases activity (Figure 3D) measured by flow cytometry. Caspase-3 activity was measured spectrophotometrically at 400 or 405 nm by the detection of p-nitroaniline (pNA) after the cleavage from the labeled substrate DEVD-pNA by the use of a microtiter plate reader [2]. MultiCaspase assay can detect the presence of active caspases-1, 3, 4, 5, 6, 7, 8, and 9 utilizing a derivatized VAD-peptide5 with a fluorescent group. The peptide is non-cytotoxic membrane permeable molecule that binds to activated caspases, the resulting fluorescent signal is proportional to active caspases [3]. Camptothecin was used as a positive control for apoptosis. More information was added to the manuscript in the Materials and Methods section pages 9 and 10, lines 271-287.
https://www.biovision.com/caspase-3-colorimetric-assay-kit.html http://www.icms.qmul.ac.uk/flowcytometry/uses/musekits/protocols/MCH100109%204600-3412MAN%20[B]%20MUSE%20MULTICASPASE%20KIT%20USER%20GUIDE.pdf What is the significance of the small cell cycle data?A: What does the reviewer mean with “small cell cycle”? It is not clear since the term “small cell cycle” was not used in the manuscript.
Cell cycle is affected in cancer cells, mainly avoiding the regulation steps that permit the genetic damage repair and undergoing to accelerated cell division. Anticancer molecules can affect cell cycle, mainly arresting cells in G0/G1 or in G2/M phases to permit genetic damage repair. Cell cycle and apoptosis are related and controlled by some regulation proteins as p53, which depends on different posttranslational modifications. When the cell suffers high level of damage, cell cycle arrest leads to induce apoptosis [4]. Our results show that TBLF induces G0/G1 arrest and early apoptosis in HT-29 cells, which suggesting that cells are undergoing to apoptosis with a previous antiproliferative effect.
https://doi.org/10.1046/j.1365-2184.2003.00266.xThank you very much for your comments.
Reviewer 2 Report
In their paper titled ‘Tepary bean (Phaseolus acutifolius) lectins induces 2 apoptosis and cell arrest in G0/G1 by p53(ser46) 3 phosphorylation in colon cancer cells’ by Moreno-Celis et al, the authors investigate if Tepary bean lectin fraction (TBLF) induces cell death in colon cancer cells and attempt to clarify the mechanism of its induction. They start from their previous findings that TBLF is suppressing growth of a colon cancer model in vivo (cited as ref. 24) by inducing apoptosis in a p53-phospho S46-dependent manner, which also includes upregulation of AKT and caspase phosphorylation. Based on their finding they conclude that TBLF induces apoptosis and not necrosis. Some of the presented evidence supports this, but the paper contains contradictions and the conclusions should be much better documented and supported. Furthermore, to expand on their previous published work in vivo, the authors should make use of the much better defined cell culture system to actually show something new, not to present again what they have already presented in vivo. The paper is not carefully prepared. In fact, the preparation is downright sloppy. Important aspects presented in the figures are not described in the text whatsoever, and important controls are not presented. Furthermore, description of figures is insufficient. Overall, the paper gives impression that it was not carefully prepared, which makes this reviewer wonder about how solid the conclusions actually are.
Tepary bean lectin fraction (TBLF) is not characterized, the material is totally undefined The x axis in figure 1 is strange. How come there is an effect already at TBLF concentration of 0? This reviewer presumes that 0 should be 1 in the logarithmic scale. Please correct or explain, this makes no sense. Furthermore, why are the dose response curves in Figure 1 not shown in the same panel? It would then be clear immediately, which cell lines is affected the most by the treatment. Indicate in axis description this is a log scale. How come cell survival is above 100% in the control situation? How was cell death in Figure 1 measured? Please specify. If that is by counting of the cell number, this will be affected not only by cell death but also by inhibition of proliferation. What is camptothecin shown if Figure 2 and further?. It is mentioned nowhere in the text and the reason for its inclusion is therefore unclear. Why is HT-29 cell line most resistant in measurements in Figure 1, while it is most sensitive (the highest percentage of total apoptosis) in Figure 2. This makes no sense to me. Is there a technical issue with the experiments? According to dot plots, RKO cell lines seems to me to have much more total cell death than HT-29. If HT-29 is most sensitive for apoptosis but least sensitive for cell death measurement in Figure 1, than this does not support the view of the authors that apoptosis is the most important mode of cell death. Furthermore, the legend says the incubation time was 24 hours, while the text states it was 8 hours. Concluding about early and late apoptosis from the data presented in Figure 2 is inappropriate. The authors made no time course measurements, which would show there is first accumulation of annexin 5-positive cells in the lower right quadrant and than the cloud of cells moves into the upper right quadrant when they also become PI-positive. This time course needs to be done before the conclusions can be made. Gating for RKO is problematic, different between control and treated groups. Figure 3: There is no LDH release probably because the incubation time with the lectin was only 8 hours. Experiments in Figure 2 where anexin 5 staining was measured were done after 24 hours (according to the figure legend). No wonder LDH was not released from cells in such a short time frame. While the authors provide evidence for caspase 3 activation in Figure 3 which definitely supports their claim, this is not yet sufficient to conclude the cell death is apoptotic (it only indicates there is an apoptotic component to it) To exclude necrosis and prove apoptosis, the authors should apply pan caspase inhibitor, such as zVAD, to make their conclusions solid. If they really wish to be proper, they should also similarly employ the necrosis inhibitor necrostatin. Figure 4: cell cycle arrest is G0/G1. How do you explain that the percentage of cells in G0/G1 is increased, but the percentage in S or G2/M is not changed? Does that mean that there are more cells than 100% in the total? Furthermore, from the bars in figure 4, this reviewer is unable to get 100% as the total sum of the cells in the cell cycle. The sum of all bars in figure 4 should be 100. Figure 5: It is not clear what the figure shows. Is it mRNA or protein? This should be clear from the axis, now it is buried somewhere towards the end of the legend. Figure 6: There is not information about total expression of proteins tested for the specific phosphorylation, so the figure as is makes little sense. The loading control is not shown. The caspase antibodies used are specific for the total caspase protein, not against phosphorylated caspase protein. In discussion they claim the phosphorylation caspases were determined: Line 201: Our results show that TBLF treatment increases p-p53 (ser46) level at 8 h, in fact it was possible to observe that caspases 8 and 9 were phosphorylated (inactivated) until the first 2 h of treatment but it remained dephosphorylated from 203 4 h onwards. If p53 is mutated in HT-29 and SW-480, how can phosphorylation at Ser46 play a role if p53 is inactive? How many times were the experiments repeated? No information is provided. The conclusions of the paper could just as easily be associations without any causal relationship. Not solid proof of causality is provided.Author Response
Comments and Suggestions for Authors
In their paper titled ‘Tepary bean (Phaseolus acutifolius) lectins induces 2 apoptosis and cell arrest in G0/G1 by p53(ser46) 3 phosphorylation in colon cancer cells’ by Moreno-Celis et al, the authors investigate if Tepary bean lectin fraction (TBLF) induces cell death in colon cancer cells and attempt to clarify the mechanism of its induction. They start from their previous findings that TBLF is suppressing growth of a colon cancer model in vivo (cited as ref. 24) by inducing apoptosis in a p53-phospho S46-dependent manner, which also includes upregulation of AKT and caspase phosphorylation. Based on their finding they conclude that TBLF induces apoptosis and not necrosis. Some of the presented evidence supports this, but the paper contains contradictions and the conclusions should be much better documented and supported. Furthermore, to expand on their previous published work in vivo, the authors should make use of the much better defined cell culture system to actually show something new, not to present again what they have already presented in vivo.A: Thank you very much, in fact we present a deeper mechanism of action of the TBLF effect on cancer cells. In the in vivowith manuscript, we presented for the first time that the TBLF was able to inhibit early tumorigenesis and that the effect was related to apoptosis induction. Here we show that TBLF is to cell cycle arrest in G1/G0 phase and that phosphorylation of p53 in serine 46 is involved in the apoptotic mechanism. We also describe that Akt pathway is activated during the first 2 h and latter deactivated leading to the possibly activation of caspases. The study of the mechanism of action in in vitrosystems is necessary to for a better understanding of the molecular effect.
The paper is not carefully prepared. In fact, the preparation is downright sloppy. Important aspects presented in the figures are not described in the text whatsoever, and important controls are not presented. Furthermore, description of figures is insufficient. Overall, the paper gives impression that it was not carefully prepared, which makes this reviewer wonder about how solid the conclusions actually are.A: Thank you for your observation. We revised very carefully the manuscript and the detected mistakes were corrected. Also a language edition review was done.
Tepary bean lectin fraction (TBLF) is not characterized, the material is totally undefinedA: TBLF has been studied andtwo lectins present in this TBLF fraction, have been characterized, we add a reference by Torres-Arteaga et al (Ref 56).
The x axis in figure 1 is strange. How come there is an effect already at TBLF concentration of 0? This reviewer presumes that 0 should be 1 in the logarithmic scale. Please correct or explain, this makes no sense.A: The “x” axis in Figure 1 was not in a logarithmic scale, it was in mg of TBLF per mL, where 0 represents the control cells without TBLF (negative control). However, the confusion may be due because we presented the equations for the calculation of the LC50for each cell line, where logarithmic curves were done but not shown. Therefore, Figure 1 was changed in order to be clear and the logarithmic graphs are presented as well as the equations.
Furthermore, why are the dose response curves in Figure 1 not shown in the same panel? It would then be clear immediately, which cell lines is affected the most by the treatment. Indicate in axis description this is a log scale. How come cell survival is above 100% in the control situation?A: Each cell line is presented in separate panel because the tested concentrations were different. As the graphs were presented before, treatment 0 were the negative control cells, all the treatments were compared to cells before the treatment (data not shown) that represents 100% of cell survival. That is the reason why negative control cells or some treatment could be higher than 100% of cell survival. However, in order to be clear, graphs were changed for the logarithmic version so we expect to show better results than before. The most sensible cells (lower LC50) were SW-480 cells and the most resistant ones were HT-29 cells with a LC50almost 10 times higher than RKO cells and 100 times higher that SW-480 cells.
How was cell death in Figure 1 measured? Please specify. If that is by counting of the cell number, this will be affected not only by cell death but also by inhibition of proliferation.A: Cell death was measured by counting cell number as explained in page 8, lines 237-244. Cells were synchronized before treatments. In order to differentiate between cell death and inhibition of cell proliferation we compared them with respect to cells before the treatments (cell survival control). After this, treatments were added a negative control was included (cell proliferation control). Therefore, when treatments are compared to the cell survival control, the result is cell death and when treatments are compared to cell proliferation control the result is inhibition of proliferation.
What is camptothecin shown if Figure 2 and further?. It is mentioned nowhere in the text and the reason for its inclusion is therefore unclear.A: Camptothecin is a positive control of apoptosis; it is mentioned in figure legends where it was used, in page 4 line 111 and 118, and in page 9 lines 251-254 we include a brief description.
Why is HT-29 cell line most resistant in measurements in Figure 1, while it is most sensitive (the highest percentage of total apoptosis) in Figure 2. This makes no sense to me. Is there a technical issue with the experiments? According to dot plots, RKO cell lines seems to me to have much more total cell death than HT-29. If HT-29 is most sensitive for apoptosis but least sensitive for cell death measurement in Figure 1, than this does not support the view of the authors that apoptosis is the most important mode of cell death. Furthermore, the legend says the incubation time was 24 hours, while the text states it was 8 hours. Concluding about early and late apoptosis from the data presented in Figure 2 is inappropriate. The authors made no time course measurements, which would show there is first accumulation of annexin 5-positive cells in the lower right quadrant and than the cloud of cells moves into the upper right quadrant when they also become PI-positive. This time course needs to be done before the conclusions can be made. Gating for RKO is problematic, different between control and treated groups.A: HT-29 cells showed the highest LC50after 24 h of treatment with the TBLF. Using this concentration, apoptosis was measured after 8 h of treatment and HT-29 cells showed the highest percent of early and total apoptosis (similar to positive control cells treated with camptothecin). However, no treated HT-29 cells showed near to 20% off basal apoptosis, taking this into account the net effect of TBLF on early apoptosis was of 20%. On the other hand, RKO cells showed the highest level of latter apoptosis (25%) but the lowest effect on early apoptosis, suggesting that TBLF exhibits an acute effect on cell death in this cell line. For SW-480 cells, level of early apoptosis was of 15% and late apoptosis of 7%, suggesting that the effect at 8 h is probably not enough since the LC50 was determined at 24 h of treatment. Our results show that RKO cells were more sensitive than HT-29 cells because apoptosis was in an advanced stage. Total apoptosis (subtracting baseline apoptosis data in control cells) was calculated and data are shown in page 3, lines 93-93
Time-course measurements were assayed in previous experiments for apoptosis and LDH releases, where 8 h seems to be the best time form the determination. The incubation time for all the experiments except for the concentration-response curves was of 8 h, it was a mistake in the figure legend of Figure 2. Thank you for the observation.
Figure 3: There is no LDH release probably because the incubation time with the lectin was only 8 hours. Experiments in Figure 2 where anexin 5 staining was measured were done after 24 hours (according to the figure legend). No wonder LDH was not released from cells in such a short time frame. While the authors provide evidence for caspase 3 activation in Figure 3 which definitely supports their claim, this is not yet sufficient to conclude the cell death is apoptotic (it only indicates there is an apoptotic component to it) To exclude necrosis and prove apoptosis, the authors should apply pan caspase inhibitor, such as zVAD, to make their conclusions solid. If they really wish to be proper, they should also similarly employ the necrosis inhibitor necrostatin.A: LDH release and apoptosis assays were done at 8 h of TBLF treatment. Other experiments show that LDH was assayed at longer times showing the same results therefore, our evidence is focusing on apoptosis induction. Thank you for the suggestions about using zVAD or necrostatin as apoptosis or necrosis inhibition, respectively. We will take them into account for the new set of further experiments, however, presently it is not possible for us to do the assays.
Figure 4: cell cycle arrest is G0/G1. How do you explain that the percentage of cells in G0/G1 is increased, but the percentage in S or G2/M is not changed? Does that mean that there are more cells than 100% in the total? Furthermore, from the bars in figure 4, this reviewer is unable to get 100% as the total sum of the cells in the cell cycle. The sum of all bars in figure 4 should be 100.A: We do not understand the idea, what does the reviewer means by saying that percentage of cells in S or G2/M not changed? Each group was evaluated in separate experiments, so percentages need to be taken in separate for each one. There are nor more than 100% of cells in any of the experiments. Figure 4 was modified for a better understanding.
Figure 5: It is not clear what the figure shows. Is it mRNA or protein? This should be clear from the axis, now it is buried somewhere towards the end of the legend.A: The figure legend indicates that it is gene expression. The “y” axis was corrected.
Figure 6: There is not information about total expression of proteins tested for the specific phosphorylation, so the figure as is makes little sense. The loading control is not shown. The caspase antibodies used are specific for the total caspase protein, not against phosphorylated caspase protein.A: We used an antibody against p-Caspase-9 antibody and for cleaved caspase-8. However, as we did not determined not cleaved caspase-8, we decided to eliminate the result for caspase-8. Figure 6 was modified and we included the loading control.
In discussion they claim the phoshorylation caspases were determined: Line 201: Our results show that TBLF treatment increases p-p53 (ser46) level at 8 h, in fact it was possible to observe that caspases 8 and 9 were phosphorylated (inactivated) until the first 2 h of treatment but it remained dephosphorylated from 203 4 h onwards. If p53 is mutated in HT-29 and SW-480, how can phosphorylation at Ser46 play a role if p53 is inactive?A: For HT-29 cells, p53 is overproduced and with a G-A mutation in codon 273 resulting in an Arg-His substitution. The one who is mutated does not mean that it is inactive, p53 mutations in HT-29 cells have shown a gain in proliferative, oncogenic function, as well as in the invasion and metastasis conduction, involving mediators such as MET and increasing the activity of tyrosine-kinase receptors, being the most studied the EGFR. It has been shown that blocking of receptors related to mutations 273H and R175H of p53 modulates the malignant effect.
How many times were the experiments repeated? No information is provided.A: The experiments were performed in triplicate in at least two independent experiments as we indicated in Materials and Methods section.
The conclusions of the paper could just as easily be associations without any causal relationship. Not solid proof of causality is provided.A: Our results confirm that the TBLF induces apoptosis in colon cancer cells and we show in this work that also cell cycle arrest is involved. Also,we found that TBLF induces p53 expression and that the protein is phosphorylated in ser46.
Thank you very much for your comments.
Reviewer 3 Report
The authors showed the differential cytotoxic effect of TBLF using three different type of colon cancer cell lines. Although Figures themselves are clearly presented overall, description is hard to understand and confused because of lack of enough information.
For example,
1. the authors described on the characteristics of these cancer cell lines in methods section, which seems very important to understand this study. Therefore, this information should be added in introduction section.
2. why especially the authors exmined the only p-p53(serine 46)? is there any sepecial reason?
Author Response
Comments and Suggestions for Authors
The authors showed the differential cytotoxic effect of TBLF using three different type of colon cancer cell lines. Although Figures themselves are clearly presented overall, description is hard to understand and confused because of lack of enough information.
For example,
the authors described on the characteristics of these cancer cell lines in methods section, which seems very important to understand this study. Therefore, this information should be added in introduction section.A: We appreciate your suggestion, but the authors consider that given the approach of the paper it is more appropriate that the description of the cell lines used would be presented in the materials and methods section.
why especially the authors exmined the only p-p53(serine 46)? is there any sepecial reason?A: There is evidence that indicates the participation of p53 as a transcription factor and as a tumor suppressor is very important in making decisions for the fate of a damaged cell, although the mechanisms by which they can be induced are not fully known [1, 2]. Such events, but specific phosphorylation have been observed in various events of death and cell survival; that trigger the specific activation of genes or regulate the permeability of the mitochondrial membrane, it has been observed that phosphorylation of p53 in ser15 and ser20 occur during slight damage to DNA; while phosphorylation in ser46 is involved in cell death [2-4]. Phosphorylation of p46 ser46 (p-p53 (ser46)) is related to genotoxic stress and occurs after several hours of damage to the cell when the cell death process could be considered irreversible; which leads to a cell cycle arrest and activation of control points [4,5]. Some natural compounds have proved the induction of ser46 in p53; such is the case of the induction of cell death induced by an extract of Zelkova serrata, which shows on the one hand arrest of the S-phase cell cycle, activation of caspase -8 and an increase in the amount of p-p53 (ser46) in oral cancer cells,in contrast to non-cancerous fibroblasts [6]. The text was added to the discussion. This explanation was added in page 9, lines 199-211
Meulmeester, E.; Jochemsen, A. p53: A Guide to Apoptosis. Cancer Drug Targets2008,8, 87–97. Feng, L.; Hollstein, M.; Xu, Y. Ser46 phosphorylation regulates p53-dependent apoptosis and replicative senescence. Cell Cycle2006, 5, 2812–2819 D’Orazi, G.; Cecchinelli, B.; Bruno, T.; Manni, I.; Higashimoto, Y.; Saito, S.; Gostissa, M.; Coen, S.; Marchetti, A.; Del Sal, G.; et al. Homeodomain-interacting protein kinase-2 phosphorylates p53 at Ser 46 and mediates apoptosis. Cell Biol.2002, 4, 11–19. Smeenk, L.; van Heeringen, S.J.; Koeppel, M.; Gilbert, B.; Janssen-Megens, E.; Stunnenberg, H.G.; Lohrum, M. Role of p53 Serine 46 in p53 Target Gene Regulation. PLoS One2011, 6, e17574. Liebl, M.C.; Hofmann, T.G. Cell Fate Regulation upon DNA Damage: p53 Serine 46 Kinases Pave the Cell Death Road. BioEssays2019, 41, 1900127. Kang, H.-J.; Jang, Y.-J. Selective apoptotic effect of Zelkova serrata twig extract on mouth epidermoid carcinoma through p53 activation. J. Oral Sci.2012,4, 78–84.Thank you very much for your comments.
Round 2
Reviewer 1 Report
The points have been addressed. Much of the English writing issues are fixed. Recommend carefully through again to groom text.
Author Response
Thank you very much for you help.
Reviewer 2 Report
The authors improved the presentation of their data to some extent, so now it is a bit more comprehensible, even though it is still by no means perfect. The paper is not particularly novel, and the authors still operate with terms like 'early' and 'late' apoptosis, which is not justified. They do not provide total expression levels for the p53 and AKT, which would be appropriate when they show the phosphorylated forms, and this experiment is therefore hard to interpret. Their conclusions about apoptosis/necrosis are iffy in the least, they do show induction of cell death with some apoptotic component. This still does not mean that apoptosis is the sole means of cell death induction. As such, these results are overinterpreted. At least they removed the most blatant inconsistencies and mistakes, such as saying that they blotted phospho caspase 9 with an antibody against total caspase 9.Author Response
1. The authors improved the presentation of their data to some extent, so now it is a bit more comprehensible, even though it is still by no means perfect. The paper is not particularly novel, and the authors still operate with terms like 'early' and 'late' apoptosis, which is not justified.
Our work shows, for the first time, information about the effect of Tepary bean lectins on the mechanism of cell death. Although there is information on the apoptotic effect of lectins, each lectin acts differently from its specificity to cell membrane carbohydrates, particularly cancer cells.
About the terms “early” and Late” apoptosis, we are considering that apoptosis can be detected by using markers for phospahtidylserine membrane translocation for early stages of apoptosis and DNA fragmentation for latter stages of apoptosis. In this case, representative dot plots by the flow cytometry analysis show phosphatidylserine detected using annexin V (“x” axis) and DNA fragmentation with, 7-Aminoactinomycin D (7-AAD) (“y” axis). Four different stages of cell viability are distinguished [1]. 1) Low signal of both, annexin V and 7-ADD means live cells (lower left panel), 2) high annexin V and low 7-ADD means early apoptosis (lower right panel), 3) high annexin V and high 7-ADD means latter apoptosis (upper right panel), and 4) low annexin V and high 7-ADD means fully dead cells, mostly nuclear debris (upper left panel); each panel was represented in the bar graphs.
http://www.icms.qmul.ac.uk/flowcytometry/uses/musekits/protocols/MCH100105%204600-3384MAN%20[B]%20%20ANNEXIN%20V%20&%20DEAD%20CELL%20KIT%20100%20TEST%20USER%27S%20GUIDE.pdf
2. They do not provide total expression levels for the p53 and AKT, which would be appropriate when they show the phosphorylated forms, and this experiment is therefore hard to interpret.
Thank you, we are only showing the phosphorylated forms of the three proteins.
3. Their conclusions about apoptosis/necrosis are iffy in the least, they do show induction of cell death with some apoptotic component. This still does not mean that apoptosis is the sole means of cell death induction. As such, these results are overinterpreted.
The necrotic effect was ruled out by determining LDH. Apoptosis was determined by activation of caspases and flow cytometry. We agree that apoptosis may not be the only mechanism of induction of cellular sample. In this case, what we can say is that there is induction of apoptosis via caspases where phosphorylation of p53 in ser46 is involved. Conclusions were modified.
4. At least they removed the most blatant inconsistencies and mistakes, such as saying that they blotted phospho caspase 9 with an antibody against total caspase 9.
If at any time it was expressed that phosphorylated caspase-9 had been determined with an antibody against total caspase-9 we apologize. It clearly cannot be and we appreciate your observation.
The manuscript was already sent for English editing.
Thank you very much for your help.
Reviewer 3 Report
What I mentioned was explained, but here is few things to modify.
Is that 1% difference in Figure 2C meaningful? is it really significant? In Figure 6, line for statistic significant difference is shifted.Author Response
We appreciate your observations, Figures 2 and 6 have been corrected.
Thank you very much for your help.